# Electrical switching of high-performance bioinspired nanocellulose nanocomposites

Dejin Jiao[1,2,3], Francisco Lossada[1,2,3], Jiaqi Guo[1,2,3], Oliver Skarsetz [1,2,3], Daniel Hoenders[1,2,3,4], Jin Liu[1,2,3] & Andreas Walther [1,2,3,4,5 ✉]

Nature fascinates with living organisms showing mechanically adaptive behavior. In contrast to gels or elastomers, it is profoundly challenging to switch mechanical properties in stiff bioinspired nanocomposites as they contain high fractions of immobile reinforcements. Here, we introduce facile electrical switching to the field of bioinspired nanocomposites, and show how the mechanical properties adapt to low direct current (DC). This is realized for renewable cellulose nanofibrils/polymer nanopapers with tailor-made interactions by deposition of thin single-walled carbon nanotube electrode layers for Joule heating. Application of DC at specific voltages translates into significant electrothermal softening via dynamization and breakage of the thermo-reversible supramolecular bonds. The altered mechanical properties are reversibly switchable in power on/power off cycles. Furthermore, we showcase electricity-adaptive patterns and reconfiguration of deformation patterns using electrode patterning techniques. The simple and generic approach opens avenues for bioinspired nanocomposites for facile application in adaptive damping and structural materials, and soft robotics.

[1] Institute for Macromolecular Chemistry, University of Freiburg, Freiburg, Germany. [2] Freiburg Materials Research Center, University of Freiburg, Freiburg, Germany. [3] Freiburg Center for Interactive Materials and Bioinspired Technologies, Georges-Köhler-Allee 105, University of Freiburg, Freiburg, Germany. [4] A3BMS Lab, Department of Chemistry, University of Mainz, Mainz, Germany. [5] Cluster of Excellence livMatS @ FIT—Freiburg Center for Interactive Materials and Bioinspired Technologies, Georges-Köhler-Allee 105, University of Freiburg, Freiburg, Germany. ✉email: andreas.walther@uni-mainz.de

The development of mechanically adaptive nanocomposites has been inspired by species of echinoderms, which share the fascinating ability to rapidly and reversibly alter the stiffness of their inner dermis when threatened[1–5]. Sea cucumbers can morph their inner dermis within seconds to endow essential survival traits. It has been proposed that the adaptive mechanical behavior is achieved by a proper control of the stress transfer through transient interactions within their hierarchical architecture composed of a soft and viscoelastic matrix which is reinforced with rigid, high-aspect ratio collagen fibrils[3,6,7]. Mimicking such capabilities to change mechanical properties on demand constitutes an important milestone to enable potential applications in adaptive materials systems that range among active dampening systems, soft robotics and tissue growth[8–10].

Prominent classes of synthetic mechanically switchable materials include thermo-, photo-, and chemo-responsive soft materials, such as hydrogels, elastomers, or semicrystalline resins[2,3,11–18]. While these materials show mechanical changes upon triggers (some exhibit viscosity/modulus changes of several orders of magnitude), the vast majority exhibit a very low stiffness (modulus in kPa–MPa range). Currently, mechanically adaptive materials with high and changeable stiffness in the GPa regime are extremely limited. For instance, bioinspired high-performance nanocomposite materials, inspired by biological load-bearing structures, are a particular material class that would strongly benefit from encoding mechanical adaptivity. These bioinspired nanocomposites aim for highly ordered hard/soft structures at high fractions of reinforcements (typically above 50 wt%) and with precisely engineered energy-dissipation mechanisms[19–23]. However, installing a programmable trigger and realizing an adaptation to external signals in such bioinspired nanocomposites is highly challenging as the adaptivity has to ultimately be provided through the soft component. This latter is only present at minor fractions (<50 wt%) and nanoconfinement conditions complicate the behavior[4,24–27].

In the aspect of triggers, electricity excels and is highly desirable, as it is easily accessible and controllable, highly penetrating, eco-friendly and thus of high relevance to real-life structural material applications. Recently, electricity-triggered changes in polymer materials present some progress, with the most notable examples dealing with dielectric soft actuators employing ultrahigh electric fields or electro-strictive elastomers[28–32]. However, electricity-induced changes in material properties, even simple on/off softening effects, are unprecedented in highly-reinforced high-performance bioinspired nanocomposites. The large body of work in bioinspired high-performance materials focuses on improvements of the static material behavior and new processing approaches[33–37]. To reach the adaptation of mechanical properties, we hypothesized that electrothermal heating (i.e., Joule or resistive heating) might be particularly appealing, as it is an electrolyte-free and low voltage driven process, and more critically, allows a control over the material properties (temperature and mechanical behavior) as a function of the energy input[38–40].

Here, we design electricity-adaptive, highly-reinforced bioinspired nanocomposites by incorporation of a rapid electrothermal energy transfer cascade allowing a reversible modulation of mechanical properties using low voltage direct current (DC). The bioinspired nanocomposites are formed by combining bio-sourced and sustainable wood-based cellulose nanofibrils (CNFs) with water-soluble, low-$T_g$ (=glass transition temperature) copolymers equipped with thermo-reversible supramolecular motifs. CNFs offer extremely high stiffness ($E = 135–145$ GPa), and are highly promising for renewable, biodegradable, and versatile functional applications[25,41–45]. The thermo-reversible supramolecularly linked polymers undergo efficient de-linking during Joule heating so as to be able to achieve large property changes. The electrothermal conversion is realized by simple deposition and spray coating of single-walled carbon nanotubes (SWNTs) on the bioinspired CNF/polymer nanocomposites. The thin SWNT layer serves as resistive heater, allowing a rapid, repeatable, and homogeneous heat generation in the bioinspired nanocomposites that is able to break the supramolecular bonds and enhance the molecular motion. This leads to significant softening, control over stress-relaxation properties, and the possibility to program the mechanical properties from stiff-to-soft with voltage supply. More importantly, we show that spatially selective application of voltage by differently connecting electrode patterns on the films leads to electro-programmable mechanical deformation patterns.

## Results

### Concepts and building blocks integrated into a material system.
Figure 1 summarizes the concept for electricity-adaptive bioinspired nanocomposites, in which the mechanical properties can be manipulated after structure formation with high spatial and

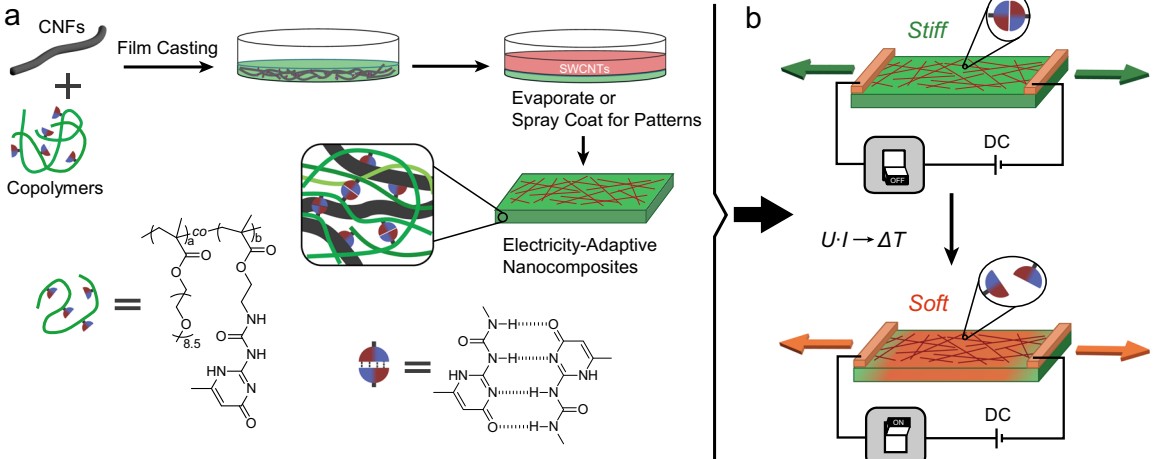

**Fig. 1 Electrical switching of bioinspired nanocomposites based on CNF and hydrogen-bonded polymers, exploiting a Joule heating to modulate thermo-reversible supramolecular bonds in the polymer binder. a** Schematic preparation of bioinspired nanocomposites by aqueous casting of CNFs and EG-UPy$_{29}$ polymers bearing fourfold hydrogen bonding dimers, and subsequent deposition of SWNTs on CNF/EG-UPy$_{29}$ nanocomposites. **b** The Joule heating leads to the dynamization and breakage of supramolecular bonds at high temperature, resulting in a macroscopic, electricity-adaptive softening.

temporal resolution using DC. It starts with the water-borne film casting toward highly-reinforced bioinspired nanocomposites based on anionic CNFs (diameter 2–4 nm; length up to 3 μm; 0.44 mmol g$^{-1}$ COOH groups; AFM in Supplementary Fig. 1) and tailor-made copolymers to form a homogeneous nanocomposite structure with a 50/50 w/w composition. The polymer phase is engineered to embed thermo-reversible hydrogen-bonded cross-links into low-$T_g$ copolymers, so that it is able to provide significant property changes upon heating by increasing the exchange of units and the decrease of the overall binding. This is the first prerequisite for realizing the electricity-adaptive properties in bioinspired nanocomposites. Another key is to locate an electrothermal conversion mechanism into the material system. This is achieved with a thin layer of COOH-functionalized SWNTs (diameter of 1–2 nm; COOH content of 3 wt%) coated onto the initial nanocomposite films, which serves as a resistive Joule heating layer when using low voltage DC. We chose a surface electrode layer rather than realizing a conductive bulk, because this gives more freedom in electrode pattern design. The overall system forms an electrothermal energy cascade converting DC to heat to provide dynamization and breakage of the thermo-reversible supramolecular bonds and to ultimately lead to macroscopic softening. We will demonstrate that even defined and programmable mechanical patterns in the bioinspired nanocomposites are achievable using selective gating of electrode patterns.

For the rational design of the polymer phase, we synthesized fully water-soluble copolymer based on poly(oligo ethylene glycol methacrylate) (EG; as a low-$T_g$ backbone) containing 29 mol% thermo-reversible fourfold hydrogen bonding motifs (ureidopyrimidinone, UPy), EG-UPy$_{29}$, Supplementary Fig. 2a–c). A pure EG homopolymer was synthesized as a reference. The polymers have a relatively low degree of polymerization and a low $T_g$ (Supplementary Fig. 2d, e, Supplementary Table 1), so that macroscopic property changes result mainly from supramolecular linkages rather than from dynamization of entanglements and $T_g$. Macroscopically, the copolymer bearing UPy motifs is an elastomer, while pure EG is a viscous melt, indicating the successful formation of hydrogen-bonded crosslinks. The supramolecular interactions mechanically stiffen the polymer as can be seen in the rheological measurements. EG-UPy$_{29}$ offers a higher storage modulus ($G'$) compared to pure EG (as seen by rheology in Supplementary Fig. 3). More importantly, EG-UPy$_{29}$ shows a pronounced solid-to-melt transition when passing beyond 62 °C (also shown in Supplementary Fig. 4), which arises from thermal dissociation and dynamization of the hydrogen-bonded UPy dimers. In contrast, pure EG does not show any transition and behaves as viscous melt within the complete temperature range (30–130 °C). The thermal transition of the polymer phase from a thermo-reversibly crosslinked network to a polymer melt is of paramount importance for the electricity-adaptive system, as it allows large changes in the mechanical properties of highly-reinforced bioinspired nanocomposites by breaking crosslinks of a nanoconfined polymer phase.

Here, we focus on nanocomposites containing CNF/polymer = 50/50 w/w, as this composition is known to reveal fundamental details on molecular engineering of bioinspired nanocomposites, and has presented a good balance between stiffness, strength, and toughness[23]. In general, higher contents of CNF reinforcements in such bioinspired nanocomposites or CNF/polymer nanopapers lead to substantial stiffening and strengthening, and a less ductile behavior as a function of the content[23,25,26]. In terms of film preparation, simple evaporation of 50/50 w/w CNF/polymer dispersions delivers homogeneous films with macroscale dimensions (typically with a diameter of 5 cm and thickness of 15 μm). After film casting, we deposited desired amounts of SWNTs on

the surface of CNF/polymer nanocomposites to form a thin film resistive heater (1–6 μm in thickness; Supplementary Fig. 5c) using a SWNT/EG-UPy$_{29}$ 90/10 w/w mixture to maintain a strong interface adhesion. Additionally, when desired, SWNT patterns can be achieved by simple, spatially selective spray coating. The thin SWNT layers produce indeed well conductive nanocomposites with sheet resistance in the range of 670–40 Ω/sq dependent on the SWNT content (3–20 wt%; relative to the total weight of the films; Supplementary Fig. 5c). These values fall within a suitable resistance range that allows a Joule heating under safe and appropriate voltages[46–48].

The inclusion of UPy motifs (29 mol%) into the polymers of the CNF/EG-UPy$_{29}$ nanocomposites leads to an increase of the Young's modulus ($E$, from 1.6 to 6.2 GPa), tensile strength ($\sigma_b$, from 53 to 107 MPa), and primary yield points ($\sigma_y$, from 28 to 62 MPa) compared to CNF/EG nanocomposites (Supplementary Fig. 5a, yield point determination in Supplementary Fig. 6), due to promoted interactions inside the polymer phase as well as at the CNF/polymer interface. The SWNT layer does not add to the mechanical behavior of CNF/polymer nanocomposites (Supplementary Fig. 5a), because of its loosely packed structure (Supplementary Fig. 5b). To further understand the temperature dependence of the supramolecular crosslinking and their corresponding mechanical response in the resulting nanocomposites, we conducted dynamic mechanical analysis (DMA) on dried CNF/polymer/SWNT (50/50/10 w/w/w) nanocomposites (Supplementary Fig. 5d). The CNF/EG-UPy$_{29}$/SWNT (50/50/10) nanocomposites show a strong loss in $G'$ at ca. 60 °C, which is reminiscent of the thermal transition from the dissociation and dynamization of UPy dimers (62 °C obtained by rheological measurements in Supplementary Fig. 3). More specifically, the $G'$ drastically decreases by 43% (from 4 to 2.3 GPa) when going from room temperature to 150 °C. In addition, the interactions may also change at the CNF/EG-UPy$_{29}$ interface, but this cannot be analyzed and decoupled spectroscopically. The situation is different for CNF/EG/SWNT (50/50/10) nanocomposites, in which the $G'$ maintains a constant behavior above room temperature, yet at a lower plateau. Critically, this confirms the importance of our molecular design strategy to embed a well-defined thermal elastomeric-to-melt transition into the polymer phase as compared to the heating of a normal CNF/EG nanocomposite, that undergoes hardly any changes. It may also be important to note, that pure CNF nanopapers with exclusive CNF/CNF interactions show hardly any temperature dependence[24].

**Joule heating as enabling effect for electrical switching**. The electrothermal effect is used to change the mechanical cohesion (dynamization and de-linking of supramolecular polymers) and accelerate the molecular motion after the structure formation using comparably low DC (Fig. 1b). To quantitatively understand the Joule heating properties of our bioinspired CNF/EG-UPy$_{29}$/SWNT nanocomposites, we applied a current along the film (size of 5.5 × 4 mm) and monitored the temperature via a forward looking infrared (FLIR) camera under varied voltage input (Fig. 2a). The average temperature rapidly increases from room temperature to ca. 105 °C as can be seen in the typical heating map series with an applied voltage of 17 V (Fig. 2b). Additionally, the Joule heating allows to steadily ascend and descend the temperature in repetitive power on/power off cycles with durations of 10 s (Fig. 2c), indicating good heating stability and repeatability. Figure 2d shows the temporal evolution of the temperature under different voltage supply (see corresponding electric current in Fig. 2e). All time-dependent heating plots follow a similar asymptotic behavior and reach a steady-state temperature within

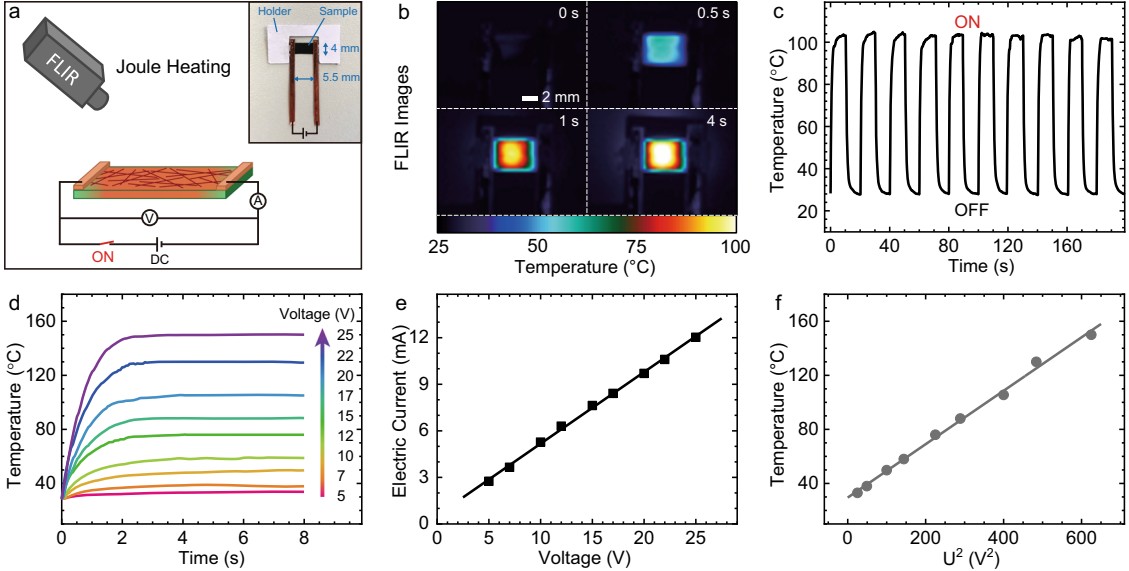

**Fig. 2 Joule heating of CNF/EG-UPy$_{29}$/SWNT (50/50/10) nanocomposites by applying DC voltage. a** Illustration of the measurement setup for Joule heating using a FLIR camera. The inset shows a photograph of the sample with copper electrodes. **b** FLIR images of the nanocomposite under applied voltage of 17 V. **c** Heating/cooling cycles under applied voltage of 17 V. **d** Time-dependent heating plots for different voltage. **e** The electric current as a function of voltage in the closed circuit. **f** The steady-state temperature as a function of square of voltage (see Eq. (1)).

only 2 s after the start, confirming a very fast response. Although the SWNTs were coated only on one side of the films a homogeneous heating is obtained throughout the entire thickness (temperatures on the backside are in Supplementary Fig. 7). A corresponding simulation of the temperature gradient (also for much thicker films) can be found in Supplementary Fig. 8. More importantly, the steady-state temperatures can be controlled by adjusting the input voltage (Fig. 2c). For example, upon applying low voltage of 5 V (2.75 mA in electric current), the steady-state temperature is only 33 °C, while when the input voltage is 25 V (11.3 mA), the temperature reaches up to ca. 150 °C. The latter temperature is clearly above the thermal transition temperature of EG-UPy$_{29}$ (Supplementary Figs. 3, 5d) and is sufficient to induce supramolecular dynamization/dissociation[49–52]. In a theoretical model, that describes the relationship between electrical and thermal properties based on the energy balance of Joule heating versus power dissipation, the temperature ($T_s$) of the steady states can be calculated by the following equation[46,53,54]:

$$T_s = T_0 + \frac{U^2}{R\alpha A} \qquad (1)$$

where $U$, $T_0$, $R$, $\alpha$, and $A$ are the supplied voltage, initial ambient temperature, resistance of the film, heat-transfer coefficient and heating area, respectively. Concretely, in the Joule heating of CNF/polymer/SWNT bioinspired nanocomposites the electric current in the circuit is proportional to the voltage (Fig. 2e), meaning a constant resistance of the thin SWNT layer at every temperature. The $\alpha$ and $A$ are also constants for certain systems[55,56]. Thus, the saturation temperature is determined solely by the applied voltage. Here, the quasi-equilibrium temperatures in steady state follows a close to linear relationship as function of the square of the voltage (Fig. 2f), indicating good agreement with the theoretical model and a facile control over the temperature by just varying the voltage. Strikingly, the establishment of tunable steady-state temperatures is important for the electricity-adaptive system, as it provides a real adaptation of mechanical properties as a function of the power strength, as we will demonstrate later.

**Adaptive mechanical behavior with electrical switching**. We investigated the electricity-adaptive mechanical properties using tensile tests and DMA at different DC voltage inputs using three important methods: 1. stress relaxation, 2. global adaptation of tensile properties at constant conditions, and 3. dynamic adaptation with varying power input. Stress relaxation provides detailed insights into the dynamization of the supramolecular bonds. To this end, the films were first stretched for 5 s (strain = 1.0 ± 0.1%, below the yield points) followed by 60 s of relaxation at constant extension, in the middle of which (from 25 to 45 s) we turned on the voltage to heat the samples for 20 s, and thereafter the films were allowed to return back to room temperature. Figure 3a shows the temporal evolutions of normalized stress during this process, which display three-step decay profiles when passing from 5 to 65 s, corresponding to the stress relaxation at ambient conditions (5–25 and 45–65 s) and with Joule heating (25–45 s). During the first halt period, the elastic stress is slightly released due to the mesoscale motion of polymer chains and potentially due to some rearrangement of UPy/UPy crosslinks under the strain condition. However, the stress relaxation is significantly accelerated once the voltage is applied and the Joule heating is triggered. Critically, the drop in stress during 20 s of Joule heating shows a temperature/power dependence (Fig. 3a, b). This indicates that the nanocomposites successfully adapt their behavior to the Joule heating. Interestingly, a critical temperature can be observed (ca. 60 °C), whereafter the stress drops more abruptly, as can be seen from the significant increase of stress decay above this temperature (Fig. 3b, c). This is directly linked to the dynamization and dissociation temperature of the Upy/Upy associates (Supplementary Fig. 3)[49,50,52]. Overall, the electricity-adaptive stress relaxation of the nanocomposites can be understood by the dynamization and breakage of the thermo-reversible crosslinks, as well as the enhanced molecular motion that occurs at higher temperatures. When the polymer phase is dynamized to be in the melt phase at high temperature, it provides nanoscale lubrication for the CNF network leading to strong relaxation. Once the temperature comes back to room temperature (after ca. 45 s), a stress increase appears as a function of the applied power

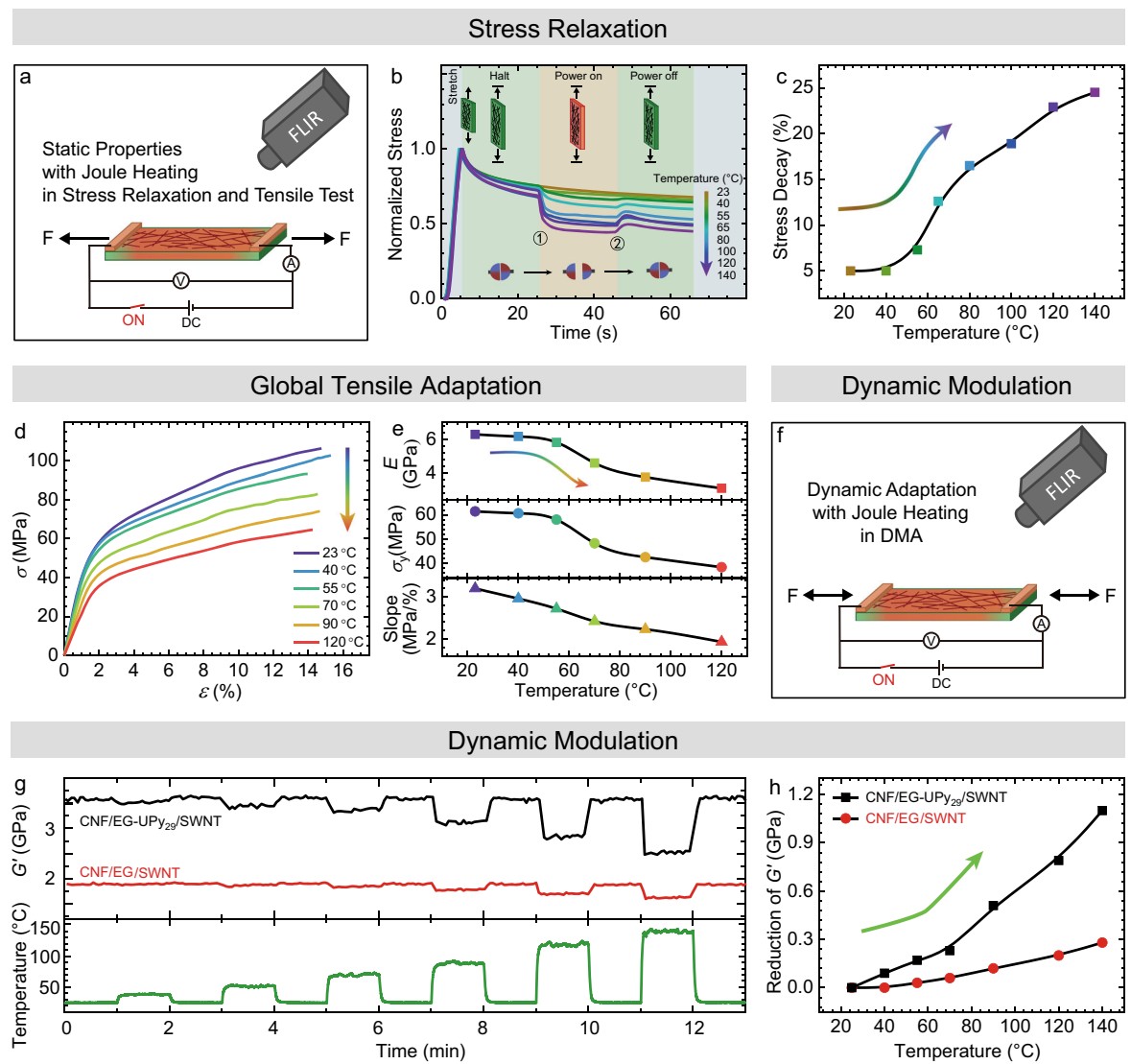

**Fig. 3 Electricity-adaptive mechanical properties in CNF/polymer/SWNT (50/50/10) nanocomposites studied under Joule heating. a** Illustration of the setup of electricity-adaptive properties testing in stress relaxation and tensile tests. The action area of the film is 5.5 × 2 mm. **b, c** Electricity-adaptive stress relaxation. **b** Temporal evolution of the stress of CNF/EG-UPy$_{29}$/SWNT (50/50/10) nanocomposites. This contains four steps: Stretching to strain = 1.0 ± 0.1%; relaxation for 20 s at ambient condition (power off); relaxation for 20 s with power on and voltage-controlled Joule heating; *power off* and further relaxation for 20 s. **c** Stress decay as a function of temperature with applied voltage from 25 to 45 s (indicated in (**a**) as from ① to ②). The stress decay defined as the relative reduction in stress during 20 s of power turn-on: $(\sigma_1 - \sigma_2)/\sigma_1$. **d, e** Electricity-adaptive global tensile properties. **d** Stress–strain curves of CNF/EG-UPy$_{29}$/SWNT (50/50/10) nanocomposites under Joule heating and varying temperature. **e** From top to bottom, the panels show the corresponding $E$, $\sigma_y$, and the slope of inelastic deformation region as a function of temperature. **f–h** Dynamic in situ modulation of the mechanical properties under Joule heating. **f** The illustration of the setup of electricity-adaptive properties testing inside a DMA using oscillatory deformation. **g** Temporal evolution of storage modulus, $G'$, tested using DMA with varied voltage to control the temperature, which shows the reversible and electricity-adaptive properties of the bioinspired nanocomposites. The black and red lines are the $G'$ for CNF/EG-UPy$_{29}$/SWNT (50/50/10) and CNF/EG/SWNT (50/50/10), and the green line the corresponds to the temperature during the tests. **h** The reduction of $G'$ as a function of temperature extracted from (**d**).

conditions, which is linked to reengagement of the hydrogen bonds and potentially thermal contraction during cooling.

Figure 3d, e depicts the electricity-adaptive tensile properties across the full tensile curve with different voltage input. In the absence of any voltage, the CNF/EG-UPy$_{29}$/SWNT (50/50/10) nanocomposite is relatively stiff and strong with high $E$ (6.2 GPa), $\sigma_b$ (107 MPa), and $\sigma_y$ (61.4 MPa)—which is well in line with typical high-performance CNF-based nanocomposites[24,25]. However, upon applying a voltage, the stiffness, tensile strength and yield strength undergo a significant decrease (Fig. 3g). For example, Joule heating the sample to 120 °C leads to an almost twofold decrease in $E$, $\sigma_b$ and $\sigma_y$ to 3.1 GPa, 65 MPa, and 38.5

MPa, respectively. This nicely confirms that the whole mechanical performance of the CNF-nanopapers experiences electricity-adaptive mechanical changes depending on the applied voltage and obtained temperature. The stiffness shows a distinct drop when the temperature overcomes 60 °C (Fig. 3e), corresponding to the dissociation regime for UPy motifs. Moreover, an abrupt reduction in $\sigma_y$ at temperature values higher than 60 °C is observed (Fig. 3e). Since the yield points are related to the onset of fibrillar sliding by CNF de-linking, it becomes obvious that this de-linking transition can also be distinctly influenced. Similarly, since the slope of the inelastic regime is linked to frictional sliding of the CNFs, we find also a decreasing slope from $\sigma_y$ to $\sigma_b$

(Fig. 3e), being indicative of managing the frictional CNF sliding in an electricity-adaptive manner.

Next we will turn to a direct and dynamic in situ manipulation of the mechanical performance by repetitive alterations of the power conditions in an in situ DMA experiment (Fig. 3f). To this end the films were stretched at a low force (0.1 N) and pre-stretched for 5 min to reach a plateau of $G'$, while afterwards applying a sinusoidal deformation ($f = 1$ Hz). During the oscillatory measurements, the films were electrically heated to vary the temperature with intervals of 1 min between each power on/power off transition (Fig. 3g). Here we also make a direct comparison between the supramolecularly linked polymer EG-UPy$_{29}$ and the pure EG polymer. One can observe that the $G'$ of the supramolecularly crosslinked CNF/EG-UPy$_{29}$/SWNT (50/50/10) nanocomposites is significantly higher than that of CNF/EG/SWNT (50/50/10), again confirming the supramolecular stiffening of the mechanical properties. Focusing on the CNF/EG-UPy$_{29}$/SWNT (50/50/10) nanocomposite, the mechanical properties show clearly a reversible electricity-induced adaptation, in which the $G'$ drops rapidly into a plateau upon the application of steady-state temperature (power on), and then fully recovers to the initial value when the sample is cooled down to room temperature (power off). This points to a striking reversibility of the adaptive mechanical properties, which can be tuned as a function of the strength of the energy input. Figure 3h shows the temperature dependence of the $G'$ at each temperature in comparison to $G'$ at room temperature. The plot shows two different regimes with a break at ca. 60 °C. After passing 60 °C, $G'$ deceases more strongly with the Joule heating, strictly relating to the dissociation and dynamization of UPy (Supplementary Figs. 3, 5d). The situation is dramatically different for CNF/EG/SWNT (50/50/10) without thermo-reversible crosslinks. Although the nanocomposite shows some reversible softening/stiffening properties in an alternative power on/power off mechanism, the reduction of $G'$ is much lower than that of CNF/EG-UPy$_{29}$/SWNT (50/50/10) (Fig. 3h) at similar temperature conditions. More specifically, $G'$ decreases only 15% (from 1.9 to 1.6 GPa) when the temperature increases from room temperature to 140 °C, whereas CNF/EG-UPy$_{29}$/SWNT (50/50/10) shows a twofold higher response. Furthermore, the reduction of $G'$ in the CNF/EG/SWNT (50/50/10) nanocomposite shows no deviation from a single linear relationship with the temperature (Fig. 3h). The performance of the CNF/EG/SWNT (50/50/10) nanocomposite corresponds to the absence of thermo-reversible crosslinks in the polymer phase, and only shows a contribution of enhancing thermal motion in a polymer melt. This control experiment hence underscores the importance of achieving thermal de-linking of the supramolecular bonds (3D network) instead of simply using a viscous polymer melt (linear chains) that can at best increase its reptational motion. Overall, the quick and reversible adaptation is remarkable for a highly-reinforced CNF/EG-UPy$_{29}$/SWNT nanocomposites and is similar to the inner dermis of the echinoderms species, that are able to rapidly and reversibly change their stiffness at their demand.

**Reconfigurable electrical switching of mechanical patterns**. Following the demonstration of homogeneous adaptation of the mechanical behavior on various elastic and inelastic and stress-relaxation aspects, we hypothesized that the correct placement and wiring of electrode patterns would also allow for an on-demand spatial control of mechanical properties in a heterogeneous and even programmable manner. Among the multitude of potential material processing method, we showcase two major scenarios for proof-of-concept purposes. Scenario 1 is achieved by changing the position of electrodes from a previous end-to-end

configuration (for global heating) to an internal wiring allowing to define a spatially selected heating zone in the center (Fig. 4a–d). Scenario 2 aims for differently wiring of a spray-coated X-electrode pattern across a sample in different configuration. (Fig. 4e–t). We show the electricity-adaptive softening during tensile tests in situ using digital image correlation (DIC) of speckle patterns to visualize the local deformation as color-coded strain maps with high spatiotemporal resolution[57–60].

In the first scenario, we used a film covered completely with a SWNT layer, but moved the electrodes from the edge of the films that we used before (Figs. 2a, 3a, f) to the middle, as shown in Fig. 4a, and fixed those in the tensile tester while applying DC on the film (scheme in Fig. 4b). In this case, only the part of the film between the electrodes receives an electric current flow (Joule heating). Figure 4c displays the corresponding strain maps for the samples at room temperature (power off) and and at different Joule heating conditions (power on). The films are divided into three parts (Fig. 4a), among which the middle part is the relevant area for studying the stiffness modulation as it undergoes Joule heating. In absence of DC (23 °C), the strains at all three parts are very similar (Supplementary Fig. 9a). Note that the strain at the electrodes is limited due to the use of copper electrodes. When turning on the voltage, the Joule heating in the centers leads to a strong focus of the deformation within the electro-adapted inner part—as a function of the applied power—providing localized on-demand softening of the material (Fig. 4c, Supplementary Movie 1). The corresponding real strains obtained from DIC images show a decrease on the non-heated parts from ca. 12 to 2%, whereas the strain for the electricity-adaptive part stays at 12–13% (Fig. 4d, Supplementary Fig. 9). This demonstration underscores that deformation in real-life application settings can be guided into appropriate areas using a simple electrical switch.

More intriguing reconfiguration of mechanical patterns can be achieved by a direct deposition of SWNTs electrode patterns using spray coating through predesigned masks (Supplementary Fig. 10). This process delivers a similarly thin SWNTs electrode films (but now as patterns) as found for the above CNF/EG-UPy$_{29}$/SWNT (50/50/10) nanocomposites prepared via two-step casting. Figure 4e–g depict the FLIR images for SWNT cross patterns while applying the voltage with different connectivity. Since the current is conducted along the shortest pathway, the heating patterns can be reconfigured on demand. This approach opens numerous possibilities for the design of synthetic heterogeneous bioinspired nanocomposites, and more advanced patterns (letters, trees and clouds) can be encoded by decorating the films with different SWNT patterns and applying voltage on them (examples in Supplementary Fig. 11).These lead to defined Joule heating regions with high fidelity and controllable temperatures, which locally softens the bioinspired nanocomposites. Remarkably, the application of a current through the patterns leads to a spatially controlled increase of the temperature, and causes a concentration of deformation in the patterns as they are wired, as shown in their strain field developments using DIC (Fig. 4k–p, Supplementary Movie 2). For example, when applying a DC on a diagonal line (generating a localized temperature of ca. 120 °C), the maximum strain for this part reaches up to ca. 8%, while the non-heated region maintains a relatively low deformation (<2%, Fig. 4k–m). The deformation pattern reconfigures, when the voltage is applied with an angle (Fig. 4n–p).

Corresponding COMSOL finite element modeling (FEM) simulations confirm the localized Joule heating induced softening to translate into programmable deformation maps (Fig. 4q, r, Supplementary Movie 2). These simulated deformation features under stretching are consistent with those observed using DIC with localized Joule heating. The possibility of FEM modeling underscores that a predictive design of tailor-made materials is possible

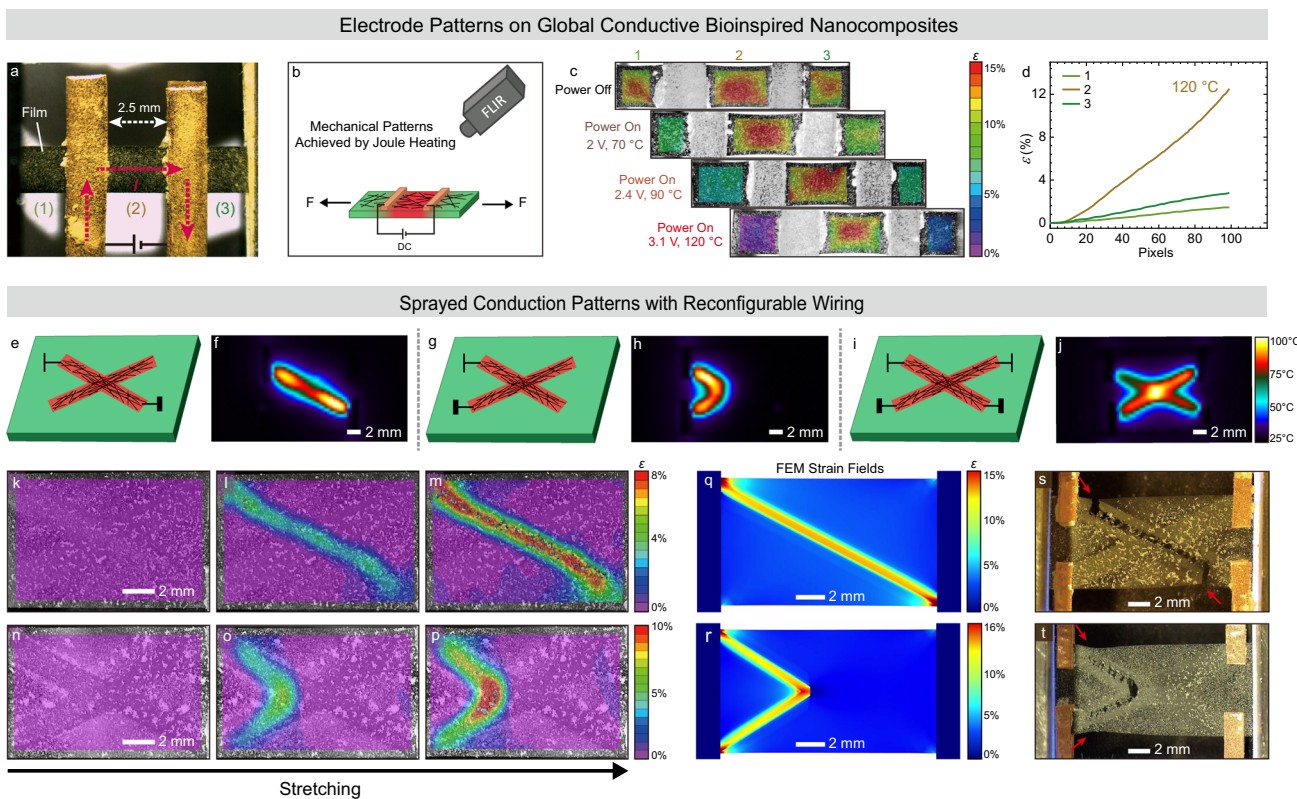

**Fig. 4 Electricity-adaptive mechanical patterns under applied voltage. a–d** Mechanical patterns achieved on a global conductive CNF/EG-UPy$_{29}$/SWNT (50/50/10) nanocomposites by changing the heating zone. **a** Photograph for the mechanical patterns in CNF/EG-UPy$_{29}$/SWNT (50/50/10) nanocomposites with copper electrodes. The distance between the copper electrodes is ca. 2.5 mm. The films are divided into three parts, while only the central part undergoes Joule heating. **b** Illustration of in situ testing setup. **c** Strain fields just before fracture at different local temperatures in the middle part (with voltage input). **d** The true strain extracted from DIC at 120 °C in the middle using DIC. The lines 1–3 represent the parts 1–3 in (**a**). **e–t** Mechanical patterns achieved via spray patterning conductive SWNT patterns with reconfigurable wiring (process in Supplementary Fig. 10). Illustrations for the films with SWNTs 2D patterns and selectively wiring and the corresponding FLIR images: (**e, f**) a diagonal, (**g, h**) an angle, and (**i, j**) a cross. Strain fields during tensile tests for differently wired patterns by applying voltage on (**k–m**) a diagonal and (**n–p**) an angle. The images in each panel are arranged from left to right for increasing global elongation, with the rightmost image just before fracture. Strain fields obtained using COMSOL FEM simulation by applying voltage on (**q**) a diagonal and (**r**) an angle. The corresponding crack behavior for the CNF/EG-UPy$_{29}$ (50/50) films with patterned Joule heating on (**s**) a diagonal, (**t**) an angle.

in the future. Interestingly, the failure of this material also changes completely and can be reconfigured on demand. Upon triggering the Joule heating in both cases, the failure is programmed to occur only at the heated regions (Fig. 4s, t). All examples for the electricity-adaptive programming of patterns demonstrate how strain energy can be localized and controllably dissipated on demand in specific sections of the nanocomposites using fast and easily controlled electricity-adaptive modulation of stiffness and yield points. As an important consequence, fracture in these nanocomposites is no longer a defect-driven process but can now be controlled by supplied voltage and restricted to selected parts.

## Discussion
We demonstrated a simple energy transfer approach for tailored, programmable and reconfigurable electricity-adaptation within highly-reinforced bioinspired CNF/polymer nanocomposites by showcasing fast, reversible electricity-adaptive modulation of mechanical properties and spatially controlled mechanical patterns upon stretching. The fabrication of those electricity-adaptive nanocomposites combines water-borne, highly-reinforced CNF nanocomposites with rationally designed copolymers bearing supramolecular motifs, and surface coated SWNT thin layers as a resistance heater. The use of thermo-reversible supramolecular

motifs is crucial for voltage-regulated adaptive reprogramming of such bioinspired nanocomposites containing large reinforcement fractions with large property changes, as they allow defined electrothermal dynamization and dissociation of crosslinks. Moreover, we showed that such electricity-adaptive mechanical properties allow lateral mechanical patterns via spatially selective softening of the nanocomposites. This can be achieved by either placing electrodes onto fully conductive films or more complex and arbitrary patterns can be obtained by spray-coating conductive flexible electrodes on the surface that can be wired up on demand to reconfigure mechanical patterns.

Compared to other spatiotemporally active molecular triggers, in particular light, we see significant advantages of electrothermal heating. For instance, direct light switching of light-responsive crosslinkers is hampered in the depth of the effect simply due to the high optical density at high fractions of active groups and the concurrent absorption of the nanocomposite material in the typical UV-blue light regime of photo-switchable motifs[45,61]. Moreover, the response dynamics are slow when such motifs are in bulk. This might be addressed partly using, e.g., more deeply penetrating NIR light, but still the challenge remains to really integrate this into an on-demand switchable materials system, as light may not easily be available in an application setting and red-shifted photo-switchable reversibly crosslinkers are only

emerging[62]. In comparison to integrating conductive material into the bulk of a bioinspired nanocomposite (e.g., graphene-based nacre-mimetics[63,64]), the approach of using surface layers is more versatile as it can also be used for nonconductive bioinspired nanocomposites as shown here. Additionally, deformation patterns and reconfiguration of deformation patterns would be hard to achieve in such approaches as they strictly require defined electrode patterns. Certainly, for increased thicknesses—which is a challenge by itself for the preparation of bioinspired nanocomposites—a bulk conductivity may need to be achieved, but clever lamination principles using electrode layers may still be able to address this challenge using the concepts developed herein, and 3D printing using conductive materials to make volume patterns may also open high opportunities[37,65–67].

Looking out to the future, the developed approach pushes sustainable mechanical high-performance materials based on cellulose nanofibers to a higher functionality level and opens ample possibilities to design mechanically adaptive high-performance materials that can be triggered by readily available low DC power supplies. The strategy should be widely applicable to other bioinspired nanocomposites. It gives rise to possibly engineer future highly-reinforced bioinspired systems with dynamic adaptation capabilities. The promotion of mechanical patterns might increase in the near future as they allow to mimic closer the structural complexity of natural materials using other precision molecular design and processing approaches, such as 3D printing. We foresee applications for adaptive damping materials, adaptive structural materials, adaptive tissue scaffolds and tissue replacements, and for soft robotics applications.

## Methods

**Materials**. Poly(ethylene glycol) methyl ether methacrylate (OEGMA, $M_n \approx$ 475 g/mol, 98%), 2-Cyano-2-propyl 4-cyanobenzodithioate (CPCBD, 98%), 2,2′-azobis (2-methylpropionitrile) (AIBN, 98%), 6-Methylisocytosine (MIC, 98%) and 2-Isocyanatoethyl methacrylate (98%) were purchased from Sigma-Aldrich. OEGMA was purified by passing through basic alumina. SWNTs with carboxyl functionalization (1–2 nm outside diameter, 3 wt% COOH groups) was purchased from SkySpring Nanomaterials.

**Preparation of cellulose nanofibrils (CNFs)**. A suspension of TEMPO-oxidized Kraft pulp oxidized under neutral conditions was set to pH = 8 with NaOH and homogenized in a LM10 Microfluidizer from Microfluidics applying four shear cycles (2 × 1400 bar, 2 × 1000 bar). The content of carboxyl groups is 0.44 mmol/g, the degree of polymerization determined by viscosimetry (DPv) is 725 and the degree of crystallinity by X-ray diffraction (XRD) is 77%.

**Synthesis of UPy-containing monomer (2-(3-(6-Methyl-4-oxo-1,4-dihydropyrimidin-2-yl)ureido)ethyl methacrylate) (UPyMA)**. MIC (4.0 g, 32.0 mmol) was added to 50 mL DMSO and heated to 170 °C for 10 min. Once the solid dissolved, the oil bath was removed. 2-isocyanatoethyl methacrylate (5.5 g, 35.0 mmol) was added immediately to the flask under vigorous stirring. The mixture was quickly cooled using a water bath. A white solid precipitated upon cooling, collected by centrifugation, washed with excess acetone for at least 4 times, and dried to obtain the pure product. $^1$H NMR (300 MHz, CDCl$_3$, δ (ppm): 12.91 (s, 1H, N$H$–C(CH$_3$)CH), 11.90 (s, 1H, N$H$–C(NH)N), 10.43 (s, 1H, CH$_2$N$H$–CO–), 6.15 (s, 1H, (CH$_3$)C=CH$H_{cis}$), 5.74 (s, 1H, NHC(CH$_3$)=C$H$CO–), 5.50 (m, 1H, (CH$_3$)C=CH$H_{trans}$), 4.2 (t, 2H, OC$H_2$), 3.50 (m, 2H, C$H_2$NH), 2.15 (s, 3H, NH–C (C$H_3$)CH), 1.89 (s, 3H, (C$H_3$)C=CH$_2$).

**Synthesis of P(OEGMA-co-UPyMA) copolymers**. OEGMA and UPyMA were copolymerized by reversible addition fragmentation transfer (RAFT) polymerization. OEGMA (2.49 g, 5.25 mmol), UPyMA (630 mg, 2.25 mmol), CPCBD (9.24 mg, 0.04 mmol), and AIBN (1.23 mg, 0.01 mmol) were dissolved in DMF (15 mL) and the mixture was degassed with N$_2$ for 30 min. Afterwards, the reaction vessel was placed into a preheated oil bath (65 °C). Samples were taken during the reaction to monitor conversion and molecular weight evolution. After certain reaction time, the reaction was stopped by cooling in an ice bath. The polymer was purified by dialysis for more than 5 days against water and subsequent freeze-drying. The molar fraction of molar percent of UPy in the final copolymer is 29 mol% as determined by $^1$H NMR. We abbreviate the copolymers with EG-UPy$_{29}$. The RAFT polymerization of OEGMA was conducted similarly, and the final polymer is abbreviated as EG.

**Preparation of nanocomposites**. The CNFs suspension (pH = 8; 0.25 wt%) was slowly added to polymers solution (0.25 wt% solution) under intense stirring until the desired weight ratio was obtained. The dispersion was stirred overnight to ensure complete homogeneity. Thereafter the suspension was placed in a centrifuge for 3 min at 3000 rpms to remove bubbles. The final CNF/polymer suspension was film casted in a 5 cm wide petri dish dried at room temperature.

In case of CNF/polymer/SWNT nanocomposites with homogeneous SWNTs thin layer, after the CNF/polymer films were dried, freshly sonicated aqueous SWNT/EG-Upy$_{29}$ (90/10 w/w, 0.05 wt%) dispersion was added into the petri dish. After drying for more than 7 days, the films were peeled off from the petri dish. The predefined SWNT patterns on the CNF/polymer nanocomposites are prepared using spray coating of SWNT/EG-Upy$_{29}$ ethanol dispersion with a mask placed on the films.

**Mechanical tests**. Tensile tests were carried out on DEBEN minitester with a 20 N load cell at room temperature. The testing speed was 0.5 mm/min. The specimens were conditioned at 20% relative humidity (RH) for more than 48 h. The specimen sizes were 10 mm × 2 mm. At least eight specimens were tested for each sample.

**Dynamic mechanical analysis**. The DMA measurements were performed using a MCR702 MultiDrive from Anton Paar with tensile loading. The test parameters were 0.05 N of preload, 0.1 N of oscillating force, and 1 Hz of frequency. The samples were dried overnight at 50 °C before testing.

**Joule heating**. The specimen of size 5.5 × 4 mm was connected into a closed circuit with a DC power supply (Basetech Bt-305). The film temperature during Joule heating was measured using a FLIR 655sc camera. The electric current was measured using a digital multimeter (Crenova MS8233D). The sheet resistance of the films was measures using a Four-Point Sheet Resistance Meter.

**Digital image correlation**. DIC was used to measure the strain fields of specimens during tensile test. This technique tracks the displacement of a random speckle pattern on a sequence of images. The strain fields with high spatiotemporal resolution were established by derivation of the displacement of the speckles against each other. In our case, the pattern was made using printer toner (black, or yellow for SWNT films) that electrostatically attaches to the surface of the film. A 0.5 megapixel camera was used to record the tests at 6 images/s using ×10 magnification. A sequence of ca. 100 images regularly spaced were extracted from the video and imported into the DIC software, VIC-2D. Each image was subdivided into elements of 33 pixels, and reference image for the correlation algorithm was actualized every five images.

**Simulation by COMSOL multiphysics**. The finite element model (FEM) simulation for the stretching of nanocomposites was carried out using the COMSOL Multiphysics software. The geometry of material in the simulation models replicates the corresponding samples under experimental testing conditions. The modulus of the heated pattern and non-heated regions were defined from our tensile tests. The samples were stretched to ε = 10%.

## Data availability

The data that support the plots within this paper and other finding of this study are available from the corresponding author upon reasonable request.

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

## Acknowledgements

We acknowledge financial support from the Volkswagen Foundation and the China Scholarship Council (CSC) for a scholarship.

## Author contributions

D.J. designed and carried out most of the experiments. F.L. helped with the preparation of CNFs and making some figures. J.G. conducted cross-sectional SEM. O.S. conducted COMSOL FEM simulations. D.H. performed the AFM measurement. J.L. helped with DMA measurements. A.W. conceived the project and co-designed some experiments. A.W. and D.J. analyzed the data and wrote the paper. All authors discussed results and commented on the paper.

## Competing interests

The authors declare no competing interests.
