## [Peer Review File · Nature Communications]

REVIEWER COMMENTS

Reviewer #1 (Remarks to the Author):

This manuscript contributed by Jiao et al. suggests a straightforward strategy to spatially control the mechanical properties of a composite material composing of cellulose nanofibrils (CNFs) as the fillers and copolymers with hydrogen bonding agent UPys as the matrix that is coated with carbon nanotubes to generate heat under direct current. The elevated temperature results in softening of the hybrid material with distinct mechanical performances such as breaking stress and stress relaxation, when compared to that at room temperature. Sophisticated control of the mechanics is achieved by using selective gating of the electrode patterns prepared by spray coating. Although embedding nano/micron wires has been used for electro-heating in various polymeric materials such as shape memory polymers and liquid crystalline elastomers/networks, combination of electro-thermal effect and supromolecular polymer materials, especially the spatially heating and programmed mechanics, is informative for the design of adaptive systems. The fabrication of the hybrid material and characterization/demonstration are completed. I think this work could be considered for publication in this journal. Other comments/suggestions are listed below.

--The tailored mechanics of the hybrid material is mainly attributed to the breaking and reformation of self-complemented UPy dimers. But in page 6 the authors mentioned the interaction at the CNF/polymer interface and CNF delinking. These interactions and the change with temperature should be characterized, which might be significant for the mechanical reinforcement of the fillers to the polymer matrix.

--Thermal-induced elastomeric-to-melt transition is mentioned at several places of the main text. If this is true, the material should experience plastic deformation during the stress relaxation test (Fig 2b). After turning off the electro-heating, the stress should not rise again. In Fig 2d the material at high temperature up to 120 oC can still maintain loading stress. These results suggest the material might be in a rubber state. This point should be considered.

--How does the content of CNF influence the mechanical properties of the hybrid materials?

--DSC is used to characterize the T_g of the copolymers, which are lower than -40 oC. But the materials are mechanically robust with extremely high Young's modulus. The state of polymer matrix should be carefully reconsidered. If it is in a soft rubber state, the hybrid material should be not so strong. I suggest extending the temperature range in DSC measurement to above 100 oC to check if there is thermal transition at ~60 oC. At room temperature the material with dense hydrogen bonds might be in a glassy state due to the presence of a large amount of UPy dimers.

--The hybrid materials in Fig 4 should experience plastic deformations. It should be meaningful if the material can restore to the original state, so that it can be repeatedly tailor the mechanical properties. Is it possible to form a lightly crosslinked network by some covalent bonds? In addition, scale bars should be added to Fig 4.

--The stress-strain curves in Fig 3d do not have clear yielding points. The method to determine the

yielding strength should be provided.

--In supplementary Fig 3, temperature sweep is carried out at a strain of 30%. Is this strain in the linear region of the material?

Reviewer #2 (Remarks to the Author):

A key question is whether this concept provides stronger effect than heating of a polymer composite based on amorphous inorganic particles, or short (<1mm) fibers, where an amorphous thermoplastic polymer matrix goes from glassy to rubbery state ?

Perhaps the nano fiber network structure and applications to films is the key, but a micro composite example may be helpful to clarify the advantage with the concept, in which way would such a material be inferior?

For the patterning, and for films, it is apparent that a nanocomposite provides advantages, but is it feasible for thicker structures? What are some estimated thickness limitations, and how could they be overcome?

There appears to be not so much materials science in this study but rather an emphasis of the "invention". Perhaps the materials science key should be the matrix. Has this concept been used before? Can the nanostructure, and mechanisms for the softening in the composite be given a stronger focus?

Referee #1: This manuscript contributed by Jiao et al. suggests a straightforward strategy to spatially control the mechanical properties of a composite material composing of cellulose nanofibrils (CNFs) as the fillers and copolymers with hydrogen bonding agent UPys as the matrix that is coated with carbon nanotubes to generate heat under direct current. The elevated temperature results in softening of the hybrid material with distinct mechanical performances such as breaking stress and stress relaxation, when compared to that at room temperature. Sophisticated control of the mechanics is achieved by using selective gating of the electrode patterns prepared by spray coating. Although embedding nano/micron wires has been used for electro-heating in various polymeric materials such as shape memory polymers and liquid crystalline elastomers/networks, combination of electro-thermal effect and supromolecular polymer materials, especially the spatially heating and programmed mechanics, is informative for the design of adaptive systems. The fabrication of the hybrid material and characterization/demonstration are completed. I think this work could be considered for publication in this journal. Other comments/suggestions are listed below.

1. The tailored mechanics of the hybrid material is mainly attributed to the breaking and reformation of self-complemented UPy dimers. But in page 6 the authors mentioned the interaction at the CNF/polymer interface and CNF delinking. These interactions and the change with temperature should be characterized, which might be significant for the mechanical reinforcement of the fillers to the polymer matrix.

In page 6, we state that the inclusion of UPy motifs leads to promoted interactions in the polymer phase as well as at the CNF/polymer interface, allowing stiffening and strengthening in mechanical properties of nanocomposites. In principle, the UPy motifs share the ability to form hydrogen bonds with the different CNF surface groups.

While we believe that the thermo-reversible de-linking is most efficient in the matrix, we cannot exclude that interactions also occur and change at the interface – this is why we point to them. However, we suggest that the bulk phase is the major player because the thermal transition in the composite coincides rather well with the transition found in pure bulk. Unfortunately, changes at the interface cannot be characterized, e.g. spectroscopically, as the abundance of different interactions leads to non-specific or non-selective bonding. We had attempted this before (eg. Using FTIR). It is also not possible to go into the direction of single fiber pullout measurements, as the CNFs are nanoscale. In summary, although the reviewer raises an interesting point, the requested data can to the best of our understanding not be obtained.

We do however not think that this is a negative point for the overall understanding of the material system concept. We added a comment that the interactions may also change at the interface, but that this can unfortunately not be analyzed (page 6).

2. Thermal-induced elastomeric-to-melt transition is mentioned at several places of the main text. If this is true, the material should experience plastic deformation during the stress relaxation test (Fig 2b). After turning off the electro-heating, the stress should not rise again. In Fig 2d the material at high temperature up to 120 oC can still maintain loading stress. These results suggest the material might be in a rubber state. This point should be considered.

The elastomeric-to-melt transition exclusively refers to the polymer (please see also new Sup. Fig. 4 for photographs); the bioinspired nanocomposite can of course not enter a melt phase due to the high fractions of reinforcements (50 wt%). We checked the position in the manuscript again and believe to be accurate in this statement.

Here we are dealing with highly-reinforced nanocomposites with 50 wt% of CNFs. The polymers are nanoconfined in the CNF network. When the polymers are molten at high temperature, the CNF network is still holding the overall structures, thus maintaining the loading stresses. The polymer melt provides nanoscale lubrication for CNF network leading to strong relaxation. Once the electro-heating is turned off, the stress increases again, which is linked to the reassociation of the hydrogen bonds and potentially thermal contraction during cooling. We added this on page 9.

3. How does the content of CNF influence the mechanical properties of the hybrid materials?

The inclusion of CNFs within the nanocomposites leads to substantial stiffening and strengthening, and a less ductile behavior. The details have been reported in previous articles and in summarized in previous reviews. (Adv. Funct. Mater.

2019, 1905309; *Acc. Chem. Res.* 2020, 2742-2748; *J. Mater. Chem. A*, 2017,5, 16003-16024). We added a sentence to the MS to guide the reader to this literature (page 6).

4. DSC is used to characterize the T_g of the copolymers, which are lower than -40 oC. But the materials are mechanically robust with extremely high Young's modulus. The state of polymer matrix should be carefully reconsidered. If it is in a soft rubber state, the hybrid material should be not so strong. I suggest extending the temperature range in DSC measurement to above 100 oC to check if there is thermal transition at ~ 60 oC. At room temperature the material with dense hydrogen bonds might be in a glassy state due to the presence of a large amount of UPy dimers.

We believe there is a misunderstanding in polymer characterization and bioinspired nanocomposite characterization.

DSC shows a T_g at -46 °C (polymer bulk); polymer bulk rheology shows a dissociation of the UPy dimers at around 62 °C (crossover of G' and G'') which is associated from the rubbery state to the melt. Note that rheology in Sup. Fig. goes to 120 °C, and would also reveal any further transitions. This is the polymer level characterization. Due to the limited bonding strength of non-flanked UPy units (not flanked by urea, so they cannot crystallize into higher ordered structures, see work by Sijbesma and Meijer), they cannot crystallize, and simple supramolecular dissociation is too weak to be observed in DSC. Mechanical characterization is more sensitive, as it measures a different observable. That is why we use these complementary techniques. As requested by the reviewer performed DSC up to a 100 °C, but the line remains of similar slope in above the T_g as expected. (in Supplementary Fig. 2e).

Due to the inherent dynamics of UPy/UPy dimers, and the use of a low T_g backbone material, the material at room temperature is an elastomer and not a glass ($T_g \ll RT$). We further added photographs to new Sup. Fig. 4. The polymer is elastomer at room temperature; when heated it turns into a melt (rheology clearly shows that).

The high Young's modulus in the composite arises from the inclusion of 50 wt% CNF in the bioinspired nanocomposites. Once in the nanocomposite state, the elastomer to melt transition of the polymer only translates into better lubrication of the CNF network and allows for easier movement and softening and toughening. The bioinspired nanocomposite cannot melt.

5. The hybrid materials in Fig 4 should experience plastic deformations. It should be meaningful if the material can restore to the original state, so that it can be repeatedly tailor the mechanical properties. Is it possible to form a lightly crosslinked network by some covalent bonds? In addition, scale bars should be added to Fig 4.

*No, this is not possible, because the bioinspired nanocomposite is not a rubber and cannot be transformed into a composite rubber at such high fractions of entangling nanoscale reinforcements (50 wt% CNF). The use of high fractions of reinforcements is an essential criterion for bioinspired nanocomposites (*Acc. Chem. Res.* 2020, 2742-2748). The plastic deformation in bioinspired nanocomposites or CNF/polymer nanopapers occurs by realignment and frictional sliding of the CNF nanofibrils, not by exclusive polymer deformation. Hence such materials based on entangling nanofibrils cannot recover the original state, to our opinion no matter what kind of molecular engineering would be done to the soft matrix. The CNF network cannot relax back to its original position after inelastic deformation and disentanglement on a slow colloidal length scale.*

We added the scale bars in Fig. 4.

6. The stress-strain curves in Fig 3d do not have clear yielding points. The method to determine the yielding strength should be provided.

We added the method in SI (Supplementary Fig. 6).

7. In supplementary Fig 3, temperature sweep is carried out at a strain of 30%. Is this strain in the linear region of the material?

Yes, the strain is in the linear region of the polymer. We added the amplitude sweep of EG-UPy₂₉ at temperature before and after thermal transition in Supplementary Fig. 3.

Referee #2:

1. A key question is whether this concept provides stronger effect than heating of a polymer composite based on amorphous inorganic particles, or short (<1mm) fibers, where an amorphous thermoplastic polymer matrix goes from glassy to rubbery state ?

*The objective of this MS is to introduce facile electrical switching to the field of **bioinspired nanocomposites**, and show **how the mechanical properties adapt to low direct current using supramolecular bonds**. The reviewer suggests an entirely different study based on micronscale composites and thermoplastic polymers.*

In general, the challenge in the bioinspired nanocomposite field is to find pathways to still allow for property changes, when only a small fraction of polymer is present, which is nanoconfined to very small dimensions due to the nanoscale fillers. In this case a calculation of the average distance of the CNFs at 50 wt% of polymer and using polymer density and CNF density at a CNF diameter of ca. 2.5 nm leads to a separation of ca. 1.5 nm (Acc. Chem. Res. 2020, 2742-2748; J. Mater. Chem. A, 2017,5, 16003-16024). In such nanoconfinement conditions, T_g , (glassy to rubbery transitions) become unclear for good materials design as the T_g is strongly influenced by the nanoconfinement. Therefore we opted for a better suited supramolecular strategy.

We do not see the point of making a random micro-composite for comparison as this is an inappropriate benchmarking, not addressing the particular design challenges in bioinspired nanocomposites.

2. Perhaps the nano fiber network structure and applications to films is the key, but a micro composite example may be helpful to clarify the advantage with the concept, in which way would such a material be inferior?

The question is difficult to understand. We address challenges of adaptive mechanical property design in bioinspired nanocomposites based on an emerging sustainably sourced nanoscale reinforcement, CNFs. This comes along with fundamental challenges of the material system design on the molecular and nanoscale. The "polymer matrix" is not a bulk material anymore as in microcomposites, which necessitates to think in new directions and understand new science. This addresses uncharted territory in general materials systems design and in understanding mechanical behavior of bioinspired nanocomposites in general.

Such films have also other advantages regarding to micro-composites than just pure mechanical design or properties, for instance, excellent gas barrier properties due to the large amount of crystalline and nanoscale CNFs, as well as transparency of the pure CNF and CNF/polymer films due the nanoscale nature.

3. For the patterning, and for films, it is apparent that a nanocomposite provides advantages, but is it feasible for thicker structures? What are some estimated thickness limitations, and how could they be overcome?

We now added the simulations on the thermal conductivity in the Supplementary Information (Supplementary Fig. 8), which show substantial bulk penetration.

To overcome the limitation of the thickness, a bulk conductivity may need to be achieved. Clever lamination principles using electrode layers may be able to address this challenge using the concepts developed herein. Additionally, the bulk could be made conductive using SWNT integration therein, and patterns in volume could be obtained by 3D printing. This was in parts already discussed in the MS in the discussion part, but we expanded this now (page 16-17).

4. There appears to be not so much materials science in this study but rather an emphasis of the "invention". Perhaps the materials science key should be the matrix. Has this concept been used before?

*Certainly, we report a **conceptual materials system design approach**. It will allow further optimization in the future*

and may serve as an inspiration for others to use similar principles for entirely different aspects such as soft robotic devices and other adaptive mechanical units. Compared to classical micro-composites, where the fibers or fillers are modified to fit to established matrix materials, the conceptual approach in bioinspired nanocomposites is different. Since the polymers cannot be treated as a simple bulk anymore (nanoconfinement), they need to be tailor-made to fit to the reinforcement that is present in high abundance and with high levels of interfaces due to nanoscale dimensions.

This is indeed the first time to introduce very simple electrical switching to bioinspired nanocomposites with GPa stiffness, in which the mechanical properties are controlled using low direct current as a function of the power intensity. Electricity is easily accessible and controllable, highly penetrating, eco-friendly and thus of high relevance to real-life structural material applications. Thus, this work provides tremendous progress in the field of mechanically adaptive or switchable bioinspired materials.

REVIEWERS' COMMENTS

Reviewer #1 (Remarks to the Author):

The authors have generally address my concerns in the response letter and the revised manuscript. I still have one question about the special thermo-adaptive mechanical properteis of the composite materials. If the polymers are in melt state, they can not sustain the applied stress effectively. The authors stated that the CNF network holds the overall network structure and maintains the loading stress. What's the interaction between the CNFs? Is this interaction influenced by temperature? These points may have been studied already. I suggest adding some description of this point to the manuscript should help readers to understand the unique behaviors of this system.

Reviewer #2 (Remarks to the Author):

I think the reponse to questions is adequate, and that the study can be published in its current form.

Referee #1: The authors have generally address my concerns in the response letter and the revised manuscript. I still have one question about the special thermo-adaptive mechanical properties of the composite materials. If the polymers are in melt state, they can not sustain the applied stress effectively. The authors stated that the CNF network holds the overall network structure and maintains the loading stress. What's the interaction between the CNFs? Is this interaction influenced by temperature? These points may have been studied already. I suggest adding some description of this point to the manuscript should help readers to understand the unique behaviors of this system.

The interactions between CNFs are hydrogen binding and physical entanglement (J. Mater. Chem. A 2017, 5, 16003-16024, Angew. Chem. Int. Ed. 2011, 50, 5438 – 5466).

These interactions in pure CNF nanopapers show hardly temperature dependence, as measured by DMA, in which the G' maintains a constant behavior until 150 °C, and at a very high plateau (Biomacromolecules 2016, 17, 2417–2426). Hence, we need the copolymer in between and we design the copolymer with thermo-reversible hydrogen bonds, so that a bigger effect can be targeted due to thermal delinking of the supramolecular bonds (3D network) instead of simply using a thermoplastic polymer (linear chains) that can at best increase its reptational motion.

We added a sentence to page 7.